# Peer review of "Improved Endurance of Ferroelectric Hf0.5Zr0.5O2 Using Laminated-Structure Interlayer"

_nanomaterials, 2023, doi:10.3390/nano13101608_

Round 1

Reviewer 1 Report

(1) The paper reports improvement of endurance and reduction of leakage current in ferroelectric Hf-Zr-O (HZO) by inserting a laminated-structure interlayer, which is interesting and timely. However, the idea of improving the endurance as well as ferroelectric properties of HZO films by the laminated structure or superlattice structure has already been proposed and has been actively discussed recently, but these previous works are not mentioned in the paper. The novlety of this paper may lie in the use of laminated-structure “interlayer”. The authors should clearly mention the advantages of this structure, compared to the fully laminated or superlattice structures. The results must be compared with the results of the fully laminated or superlattice structures, too. The reviewer thinks more systematic study on the effects of “interlayer” would be necessary to increase the value of paper.

(2) Important references on laminated structure of HfO2/ZrO2 are missing. For instance,

Y. Peng, et al., IEEE Electron Device Lett., 43, 216 (2022).

Z. Gong et al., Appl. Phys. Lett., 121, 242902 (2022)

S. L. Weeks et al., ACS Appl. Mater. Interfaces, 9, 13440 (2017).

S. Migita et al., Appl Phys. Express, 14, 051006 (2021).

Y. K. Liang, IEEE Electron Device Lett., 43, 1451 (2022).

D. Lehninger et al., Adv. Phys. Res., (2023).

(3) Mechanism for the observed improvement by the “interlayer” should be discussed clearly. Discussion part is confusing. It is mentioned that the phase transition between the T phase and M phase is more likely to occur in the “ZrO2” layer. However, it is mentioned in the next sentence that the proportion of M phase in the “HfO2” layer increases, which seems to contradict the previous comment. It is also mentioned that the influence of M phase fraction can be effectively reduced because the thickness of the laminated is reduced to 2 nm. If so, the fully laminated structure or superlattice structure would be better to maintain the ferroelectric phase. Please mention about the effect of “interlayer”. In addition, the authors claim the interlayer may cause damage to the uniformity and coherence, Please be more specific about the damage. Please show some evidence if possible.

(4) Figure 4 shows clear improvement for HZO with laminated interlayer. It would be nice to give information about sample-to-sample variation in the endurance properties.

(5) The term “remnant polarization” is used in the paper, which must be the correct term “remanent polarization”.

Author Response

Manuscript ID: nanomaterials-2369888

Authors’ responses to the reviewer’s comments

Reply to Reviewers’ comments on the manuscript submitted to nanomaterials-2369888, entitled “Improved Endurance of Ferroelectric Hf0.5Zr0.5O2 Using Laminated-Structure Interlayer” by Meiwen Chen et al. (nanomaterials-2369888). We are sincerely grateful to all reviewers, and their constructive comments will guide us to improve the quality of our manuscript. Based on the comments received from reviewers, our group had several in-depth discussions, and have significantly revised the paper according to the constructive comments. The detailed point-by-point responses to the reviewers’ comments are summarized as follows. All the revised text, data, and notes in the manuscript are highlighted in yellow.

===========================

Comments from Reviewer 1#

Comment 1-1: The paper reports improvement of endurance and reduction of leakage current in ferroelectric Hf-Zr-O (HZO) by inserting a laminated-structure interlayer, which is interesting and timely. However, the idea of improving the endurance as well as ferroelectric properties of HZO films by the laminated structure or superlattice structure has already been proposed and has been actively discussed recently, but these previous works are not mentioned in the paper.

Our response: Thank you for your comments. According to your comment, we add the research related to laminated structure and superlattice structure in the introduction.

Corresponding change in manuscript: Yes

Location of Change:

Section: Introduction

Page 2, paragraph2Recently, some researchers have focused on the study of thin film deposition methods to adjust the properties of devices. In 2016, Lu et al. reported the induced ferroelectricity of ZrO2/HfO2 bilayer, which provided the basis for the emergence of the laminated structure[1]. Subsequently, Weeks et al. reported a laminated structure (1 nm HfO2/1 nm ZrO2)×4 which shows the promising ferroelectricity[2]. However, the endurance properties have been less studied in these previous reports In 2022, Liang et al. studied the performance of superlattice structure composed of ZrO2/HfO2 at different annealing temperatures using ZrO2 layer as the starting layer and reached high endurance performance but low remnant polarization[3]. In 2023, Lehninger et al. studied the performance of HfO2/ZrO2 superlattices, which showed obvious wake up effect, low coercive field, enhanced polarization and high temperature reliability. But the endurance reached 107 cycles[4]. These previous studies have demonstrated the potential of laminated structure and superlattice structure for improving the ferroelectric and endurance performance.

--------------------------

[1] Lu, Y.; Shieh, J.; Tsai, F. Induction of ferroelectricity in nanoscale ZrO2/HfO2 bilayer thin films on Pt/Ti/SiO2/Si substrates. 2016, 115, 68-75.

[2] Weeks, S.L.; Pal, A.; Narasimhan, V.K.; Littau, K.A.; Chiang, T.; Interfaces. Engineering of ferroelectric HfO2–ZrO2 nanolaminates. 2017, 9, 13440-13447.

[3] Liang, Y.-K.; Li, W.-L.; Wang, Y.-J.; Peng, L.-C.; Lu, C.C.; Huang, H.-Y.; Yeong, S.H.; Lin, Y.-M.; Chu, Y.-H.; Chang, E.-Y. ZrO 2-HfO 2 Superlattice Ferroelectric Capacitors With Optimized Annealing to Achieve Extremely High Polarization Stability. 2022, 43, 1451-1454.

[4] Lehninger, D.; Prabhu, A.; Sünbül, A.; Ali, T.; Schöne, F.; Kämpfe, T.; Biedermann, K.; Roy, L.; Seidel, K.; Lederer, M.J.A.P.R. Ferroelectric [HfO2/ZrO2] Superlattices with Enhanced Polarization, Tailored Coercive Field, and Improved High Temperature Reliability.

Comment 1-2:The novelty of this paper may lie in the use of laminated-structure “interlayer”. The authors should clearly mention the advantages of this structure, compared to the fully laminated or superlattice structures. The results must be compared with the results of the fully laminated or superlattice structures, too. The reviewer thinks more systematic study on the effects of “interlayer” would be necessary to increase the value of paper.

Our response: Thank you very much for your suggestion. According to your comment, we fabricate the fully laminated structure device with ZrO2/HfO2=5/5. The schematic diagram of the HZO 5/5 is shown in Figure R1(b). And the electrical results of the fully laminated structure HZO 5/5 are tested, as shown in  Figure R2. Compared with the fully laminated structure, the laminated-structure interlayer  have less impact on Pr and maintain the advantage of endurance. And these data are added in the manuscript.

Figure R1. Schematic diagrams of (a) the capacitor HZO1/1, (b) the capacitor HZO5/5 and (c) the capacitor HZO1/5/1.

Figure R2 (a) P-V loops, (b) εr-V curves, (c) initial J-V curves of HZO1/1, HZO5/5 and HZO1/5/1 capacitor. The endurance characteristics of (d) leakage currents and (e) remanent polarization under cycling pulse with amplitude of ±3V at 0.1MHz of HZO1/1, HZO5/5 and HZO1/5/1 capacitor.

Corresponding change in manuscript: Yes

Location of Change:

Section: Introduction and Results

Page 4,5.

--------------------------

Comment 2: Important references on laminated structure of HfO2/ZrO2 are missing.

Our response: Thanks for your kindly remind. Relevant references have been supplemented.

Corresponding change in manuscript: Yes

Location of Change:

Section: References

Page 2, paragraph2: Recently, some researchers have focused on the study of thin film deposition methods to adjust the properties of devices. In 2016, Lu et al. reported the induced ferroelectricity of ZrO2/HfO2 bilayer, which provided the basis for the emergence of the laminated structure[1]. Subsequently, Weeks et al. reported a laminated structure (1 nm HfO2/1 nm ZrO2)×4 which shows the promising ferroelectricity[2]. However, the endurance properties have been less studied in these previous reports In 2022, Liang et al. studied the performance of superlattice structure composed of ZrO2/HfO2 at different annealing temperatures using ZrO2 layer as the starting layer and reached high endurance performance but low remnant polarization[3]. In 2023, Lehninger et al. studied the performance of HfO2/ZrO2 superlattices, which showed obvious wake up effect, low coercive field, enhanced polarization and high temperature reliability. But the endurance reached 107 cycles[4]. These previous studies have investigated the endurance characteristics of superlattice structure, but have not well balanced the relationship between endurance and remanent polarization.

Page 7, paragraph1The leakage current and the breakdown behavior are attributed to the migration and aggregation of defects. Oxygen vacancy is generally regarded as the dominant defect in ferroelectric capacitors. It is reported that the migration barriers from interfaces in HZO and laminated structure are different[5]. The vertical migration of oxygen vacancies can be suppressed in laminated structure due to the higher energy barriers. For HZO5/5 and HZO1/5/1, since the migration of oxygen vacancies is suppressed in laminated-structure, the increasement of leakage current is suppressed, resulting in improved endurance.

--------------------------

[1] Lu, Y.; Shieh, J.; Tsai, F. Induction of ferroelectricity in nanoscale ZrO2/HfO2 bilayer thin films on Pt/Ti/SiO2/Si substrates. 2016, 115, 68-75.

[2] Weeks, S.L.; Pal, A.; Narasimhan, V.K.; Littau, K.A.; Chiang, T.; Interfaces. Engineering of ferroelectric HfO2-ZrO2 nanolaminates. 2017, 9, 13440-13447.

[3] Liang, Y.K.; Li, W.L.; Wang, Y.J.; Peng, L.C.; Lu, C.C.; Huang, H.Y.; Yeong, S.H.; Lin, Y.M.; Chu, Y.H.; Chang, E.Y. ZrO2-HfO2 Superlattice Ferroelectric Capacitors with Optimized Annealing to Achieve Extremely High Polarization Stability. 2022, 43, 1451-1454.

[4] Lehninger, D.; Prabhu, A.; Sünbül, A.; Ali, T.; Schöne, F.; Kämpfe, T.; Biedermann, K.; Roy, L.; Seidel, K.; Lederer, M. Ferroelectric [HfO2/ZrO2] Superlattices with Enhanced Polarization, Tailored Coercive Field, and Improved High Temperature Reliability.

[5] Gong, Z.; Chen, J.; Peng, Y.; Liu, Y.; Yu, X.; Han, G. Physical origin of the endurance improvement for HfO2-ZrO2 superlattice ferroelectric film. 2022, 121, 242901.

Comment 3: Mechanism for the observed improvement by the “interlayer” should be discussed clearly. Discussion part is confusing. It is mentioned that the phase transition between the T phase and M phase is more likely to occur in the “ZrO2” layer. However, it is mentioned in the next sentence that the proportion of M phase in the “HfO2” layer increases, which seems to contradict the previous comment. It is also mentioned that the influence of M phase fraction can be effectively reduced because the thickness of the laminated is reduced to 2 nm. If so, the fully laminated structure or superlattice structure would be better to maintain the ferroelectric phase. Please mention about the effect of “interlayer”. In addition, the authors claim the interlayer may cause damage to the uniformity and coherence. Please be more specific about the damage. Please show some evidence if possible.

Our response: Thank you for your comments. The contradictory discussion you pointed out may be due to an error in the formulation. And we modified it as follows: “Previous studies have reported that Hf0.5Zr0.5O2 exhibits optimal ferroelectricity[1]. In the laminated structure, only part of HfO2 and ZrO2 are completely miscible at the ZrO2/HfO2 interfaces, and Hf0.5Zr0.5O2 is formed. The incompletely miscible parts of HfO2 and ZrO2 are shown as less Zr doping in HfO2 and more Zr doping in ZrO2, respectively. The more M-phase fraction is shown in less Zr-doped HfO2 films, and the more T-phase fraction is shown in rich Zr-doped HfO2 films[2]. The O-phase fraction decrease because of the increasement of T phase or M phase in incompletely miscible parts, which leads to the reduction of the remanent polarization in laminated structure. Therefore, the influence of remanent polarization can be effectively reduced in HZO1/5/1 compared with fully laminated structure HZO5/5 because the thickness of the laminate structure is small, only 1/6 thickness of the whole film.”

Corresponding change in manuscript: Yes

Location of Change:

Section: Discussion

Page 7, line 230-241.

--------------------------

[1] Kim, S.J.; Mohan, J.; Summerfelt, S.R.; Kim, J. Ferroelectric Hf0.5Zr0.5O2 thin films: A review of recent advances. 2019, 71, 246-255.

[2] Muller, J.; Boscke, T.S.; Schroder, U.; Mueller, S.; Brauhaus, D.; Bottger, U.; Frey, L.; Mikolajick, T. Ferroelectricity in Simple Binary ZrO2 and HfO2. Nano Lett 2012, 12, 4318-4323, doi:10.1021/nl302049k.

Comment 4: Figure 4 shows clear improvement for HZO with laminated interlayer. It would be nice to give information about sample-to-sample variation in the endurance properties.

Our response: Thanks for your comments. In order to show the sample-to-sample variation, we tested 10 samples of HZO1/1 and HZO1/5/1 and added this chart to the article and described it. Figure R3 shows the statistical chart of the maximum endurance and Pr max characteristics. In order to display the sample-to-sample variation, ten measurements are taken for each box. The range of the maximum cycles for endurance is 7×105 to 5×106 for HZO1/1 and 108 to 5×108 for HZO1/5/1. The endurance can be enhanced by more than two orders of magnitude in HZO1/5/1 due to the laminated-structure interlayer. The range of the Pr max value is 22.3μC/cm2 to 25μC/cm2 for HZO1/1 and 18μC/cm2 to 21.9μC/cm2 for HZO1/5/1. The high remanent polarization is exhibited in HZO1/1 and HZO1/5/1 capacitors.

Figure R3. Box chart for endurance and Pr max characteristics of HZO1/1 and HZO1/5/1 sample-to-sample variation. Each box plot consists of ten measurements.

Corresponding change in manuscript: Yes

Location of Change:

Section: Results

Page 5, line 167, paragraph1

Comment 5: The term “remnant polarization” is used in the paper, which must be the correct term “remanent polarization”.

Our response: Thanks for your comments. We have modified the term “remnant polarization” into “remanent polarization”

Corresponding change in manuscript: Yes

Location of Change:

Section: Introduction, Results and discussion

Page 1, line 17/Page 2, line 53, 63, 72, 76, 79/ Page 4, line 133, 134, 135, 155/Page 5, line 163, 172, 181, 189/Page 6, line 223, 224/Page 7, line 230, 238, 239/Page 8, line 277.

--------------------------

Reviewer 2 Report

The authors reported the enhanced endurance characteristic of the TiN/HZO/TiN ferroelectric capacitor by the 2-nm laminated structure interlayer embedded. Furthermore, the interlayered HZO devices showed good reliability of electrical device performance with a feasible strategy. Even though the authors proved the enhanced electrical characteristics and reliability, the direct/indirect experimental evidence or analytical investigation for the interlayer effect as a feasible way to enhance the reliability is not fully supported. Consequently, the following major comments and questions must be addressed in the revised paper to publish this manuscript in Nanomaterials.

1. the full name of the "TDDB" test on line 130 should be described at the beginning of the paper, both in the text and in the abstract.

2. Please include the chemical name of the precursor, such as Hf[N(C2H5)CH3]4 or Zr[N(C2H5)CH3]4 on line 63.

3. In line 73, the authors used a 30-second heat treatment at 500 degrees. But most importantly, the most important factor to tune the HZO crystalline phase is the annealing (programmed) rate. Please specify the specific heat treatment conditions.

4. It is unclear what the TEM image in Figure 1 shows. At the very least, a precise analysis of the crystalline phases (Tetragonal, Orthorhombic, Monoclinic) and crystal orientation formed in the HZO layer is required. It is also not clear from this material whether the lattice arrangement is regular or incoherent lattices are formed. More enlarged images and friendly indexing are needed.

5. Electrical measurements have confirmed the role of the 2 nm-thick laminated-structure interlayer in inhibiting the conduction current path. Is there a rationale for choosing 2 nm? What happens to device performance with interlayer thicknesses greater than 2 nm?

6. In Figure 5(e), the ten-year lifetime of the two capacitors was extracted from the Weibull plot. A formula for the lifetime prediction and a friendly explanation of this result are essential, and further explanation is needed to ensure that the inferred value is reliable.

7. In line 163, it was mentioned that the effect of M phase fraction was effectively reduced due to the laminated-structure interlayer. However, there is a lack of evidence related to M phase fraction in this study. Experimental evidence indicating a change in the crystalline phase fraction through qualitative or quantitative changes via XRD is needed.

8. There is no analytical result or evidence for oxygen vacancy in line 169. The question of whether the interlayer has a direct effect on oxygen vacancy requires experimental evidence. For example, clear evidence of actual oxygen vacancy through analysis such as an XPS depth profile is strongly recommended.

Minor typos and symbols need to be checked and corrected.

Author Response

Manuscript ID: nanomaterials-2369888

Authors’ responses to the reviewer’s comments

Reply to Reviewers’ comments on the manuscript submitted to nanomaterials-2369888, entitled “Improved Endurance of Ferroelectric Hf0.5Zr0.5O2 Using Laminated-Structure Interlayer” by Meiwen Chen et al. (nanomaterials-2369888). We are sincerely grateful to all reviewers, and their constructive comments will guide us to improve the quality of our manuscript. Based on the comments received from reviewers, our group had several in-depth discussions, and have significantly revised the paper according to the constructive comments. The detailed point-by-point responses to the reviewers’ comments are summarized as follows. All the revised text, data, and notes in the manuscript are highlighted in yellow.

===========================

Comments from Reviewer 2#

Comment 1: The full name of the "TDDB" test on line 130 should be described at the beginning of the paper, both in the text and in the abstract.

Our response: Thanks for your comments. We have added the full name of the “TDDB” in the text and in the abstract.

Corresponding change in manuscript: Yes

Location of Change:

Section: Abstract, Results and Conclusions

Page 1, line 22/Page 2, line 83/Page 8, line288.

Comment 2: Please include the chemical name of the precursor, such as Hf[N(C2H5)CH3]4 or Zr[N(C2H5)CH3]4 on line 63.

Our response: Thanks for your comments. We have included the chemical name of the precursor. “The Hf0.5Zr0.5O2 ferroelectric thin films are deposited by ALD at 280℃ using Hf[N(C2H5)CH3]4 (Tetrakis(ethylmethylamido) hafnium, TEMAHf), Zr[N(C2H5)CH3]4 (tetrakis(ethylmethylamido) zirconium, TEMAZr) and H2O as Hf precursor, Zr precursor and oxygen source, respectively.”

Corresponding change in manuscript: Yes

Location of Change:

Section: Materials and Methods

Page 2, line 91, 92.

Comment 3: In line 73, the authors used a 30-second heat treatment at 500 degrees. But most importantly, the most important factor to tune the HZO crystalline phase is the annealing (programmed) rate. Please specify the specific heat treatment conditions.

Our response: Thanks for your comments. We have added the specific heat treatment conditions. “Finally, all the samples are annealed by rapid thermal annealing (RTA) in N2 atmosphere. The annealing ramp up rate and ramp up time are 8.4℃/s and 60s. And the hold temperature and hold time are 500℃ and 30s.”

Corresponding change in manuscript: Yes

Location of Change:

Section: Materials and Methods

Page 3, line 108.

Comment 4: It is unclear what the TEM image in Figure 1 shows. At the very least, a precise analysis of the crystalline phases (Tetragonal, Orthorhombic, Monoclinic) and crystal orientation formed in the HZO layer is required. It is also not clear from this material whether the lattice arrangement is regular or incoherent lattices are formed. More enlarged images and friendly indexing are needed.

Our response: Thanks for your comments. Two messages are presented by the TEM images, one is the film thickness of HZO1/1 and HZO1/5/1, and the film thickness obtained by different deposition cycles is consistent. The other is the light and dark stratification phenomenon in the laminated-structure interlayer of HZO1/5/1, which reflects the difference between the deposition method of HZO1/1 and HZO1/5/1. However, because the thickness of the laminated-structure interlayer is only 2nm, a total of 4 layers of ZrO2 and HfO2 are deposited, which cannot be clearly exhibited. For fully laminated structure and superlattice structure, the boundary between layers can only be exhibited when both the HfO2 layer and the ZrO2 layer are greater than 1nm[1]. In the TEM image in this article, there are many grain coincidences, so the O/T/M phase cannot be accurately analyzed using the Fourier transform. In order to obtain more information about the O/T/M phase, we need to prepare more samples and need more detailed STEM experiments. We will conduct this experiment in the next work.

Figure R1. HAADF image of the 4(H10Z10) sample[1]

Corresponding change in manuscript: No

--------------------------

[1] Park, M.H.; Kim, H.J.; Lee, G.; Park, J.; Lee, Y.H.; Kim, Y.J.; Moon, T.; Kim, K.D.; Hyun, S.D.; Park, H.W. A comprehensive study on the mechanism of ferroelectric phase formation in hafnia-zirconia nanolaminates and superlattices. 2019, 6, 041403.

Comment 5: Electrical measurements have confirmed the role of the 2 nm-thick laminated-structure interlayer in inhibiting the conduction current path. Is there a rationale for choosing 2 nm? What happens to device performance with interlayer thicknesses greater than 2 nm?

Our response: Thanks for your comments. Except for the HZO1/5/1, the other two types of TiN/HZO/TiN capacitors are fabricated, HZO5/5 and HZO1/15/1. HZO5/5 is the fully laminated structure. The endurance is improved, but the remanent polarization is significantly reduced, as shown in Figure R2. For HZO1/15/1, the laminated-structure interlayer is composed of 15 cycles of ZrO2 and 15 cycles of HfO2, which also shows improved endurance but low remanent polarization, as shown in Figure R3. Comprehensively, the 2nm-thick ZrO2/HfO2=5/5 laminated-structure interlayer is the best choice that can balance the remanent polarization and endurance. In order to make the structure of the article more compact, the data of HZO1/15/1 is not presented in the article.

Figure R2 (a) P-V loops, (b) εr-V curves, (c) initial J-V curves of HZO1/1, HZO5/5 and HZO1/5/1 capacitor. The endurance characteristics of (d) leakage currents and (e) remanent polarization under cycling pulse with amplitude of ±3V at 0.1MHz of HZO1/1, HZO5/5 and HZO1/5/1 capacitor.

Figure R3 (a) P-V loops, (b) C-V curves, (c) initial J-V curves of HZO1/1, HZO5/5 and HZO1/5/1 capacitor. The endurance characteristics of (d) leakage currents and (e) remanent polarization under cycling pulse with amplitude of ±3V at 0.1MHz of HZO1/1, HZO1/5/1 and HZO1/15/1 capacitor.

Corresponding change in manuscript: No

Comment 6: In Figure 5(e), the ten-year lifetime of the two capacitors was extracted from the Weibull plot. A formula for the lifetime prediction and a friendly explanation of this result are essential, and further explanation is needed to ensure that the inferred value is reliable.

Our response: Thanks for your comments. The Weibull plot is applied with cumulative density probability, and the maxi-mum likelihood method is used to fit the data under DC stress, as shown in Figure 5(c) and (d). The cumulative density function in the Weibull plot is given by:

W(x)=Ln(-Ln(1-F(x)))=βLn(x/α)                                  (1)

where the x is the TBD, α is the scale-factor and β is the shape-factor.

The relationship between operating voltage and lifetime can be obtained by the statistics of 63.2% failure points in the Weibull distribution, and linearly fitting and extrapolating the DC voltage under the failure point.

Corresponding change in manuscript: Yes

Location of Change:

Section: Results

Page 5-6, line 198-205.

Comment 7: In line 163, it was mentioned that the effect of M phase fraction was effectively reduced due to the laminated-structure interlayer. However, there is a lack of evidence related to M phase fraction in this study. Experimental evidence indicating a change in the crystalline phase fraction through qualitative or quantitative changes via XRD is needed.

Our response: Thanks for your comments. There may be some misrepresentation in the discussion of the mechanism, we are very sorry for the misunderstanding. “Previous studies have reported that Hf0.5Zr0.5O2 exhibits optimal ferroelectricity[1]. In the laminated structure, only part of HfO2 and ZrO2 are completely miscible at the ZrO2/HfO2 interfaces, and Hf0.5Zr0.5O2 is formed. The incompletely miscible parts of HfO2 and ZrO2 are shown as less Zr doping in HfO2 and more Zr doping in ZrO2, respectively. The more M-phase fraction is shown in less Zr-doped HfO2 films, and the more T-phase fraction is shown in rich Zr-doped HfO2 films[2]. The O-phase fraction decrease because of the increasement of T phase or M phase in incompletely miscible parts, which leads to the reduction of the remanent polarization in laminated structure. Therefore, the influence of remanent polarization can be effectively reduced in HZO1/5/1 compared with fully laminated structure HZO5/5 because the thickness of the laminate structure is small, only 1/6 thickness of the whole film.” We did XRD for HZO1/1 and HZO1/5/1 samples, as shown in Figure R4. Due to the small difference of the Pr value between the HZO1/1 and HZO1/5/1 capacitors, XRD plots do not particularly clearly show the difference of the O phase and the M phase intensity between the two capacitors.

Figure R4. XRD images of HZO1/1 and HZO1/5/1 capacitors.

Corresponding change in manuscript: No

--------------------------

[1] Kim, S.J.; Mohan, J.; Summerfelt, S.R.; Kim, J. Ferroelectric Hf0.5Zr0.5O2 thin films: A review of recent advances. 2019, 71, 246-255.

[2] Muller, J.; Boscke, T.S.; Schroder, U.; Mueller, S.; Brauhaus, D.; Bottger, U.; Frey, L.; Mikolajick, T. Ferroelectricity in Simple Binary ZrO2 and HfO2. Nano Lett 2012, 12, 4318-4323, doi:10.1021/nl302049k.

Comment 8: There is no analytical result or evidence for oxygen vacancy in line 169. The question of whether the interlayer has a direct effect on oxygen vacancy requires experimental evidence. For example, clear evidence of actual oxygen vacancy through analysis such as an XPS depth profile is strongly recommended.

Our response: Thanks for your comments. XPS is indeed a good characterization method for testing oxygen vacancies. But in this article, the oxygen flux in the process of HZO1/1 and HZO1/5/1 capacitors are the same, so there is no difference in the number of oxygen atoms. We believe the oxygen vacancies distribution distinction in HZO1/1 and HZO1/5/1 (aggregation or diffusion) are response for the endurance difference. Unfortunately, XPS cannot clearly confirm the distribution of oxygen vacancies. And characterization of the distribution of oxygen vacancies is a challenge, especially in a film as thin as 2nm. So, we have cited some relevant references to analyze the function of oxygen vacancies in ferroelectric films and laminated structure[1-3]. To understand the different distribution of oxygen vacancies in ZrO2 and HfO2, Bai et al. carried out the first principles calculations on the energetics of filament phase segregation in HfO2 and ZrO2, revealing that oxygen vacancies are more likely to aggregation in HfO2[3]. We hope to complete the characterization of the oxygen vacancy distribution in the films in the next work.

Corresponding change in manuscript: Yes

Location of Change:

Section: Discussion

Page 7, line 235-238: Oxygen vacancy is generally regarded as the dominant defect in ferroelectric capacitors, which play an important role in the formation of the ferroelectric phase, but high concentration results in the degradation of device performance[1-3].

--------------------------

[1] Zhou, Y.; Zhang, Y.; Yang, Q.; Jiang, J.; Fan, P.; Liao, M.; Zhou, Y. The effects of oxygen vacancies on ferroelectric phase transition of HfO2-based thin film from first-principle. 2019, 167, 143-150.

[2] Glinchuk, M.D.; Morozovska, A.N.; Lukowiak, A.; Stręk, W.; Silibin, M.V.; Karpinsky, D.V.; Kim, Y.; Kalinin, S.V. Possible electrochemical origin of ferroelectricity in HfO2 thin films. 2020, 830, 153628.

[3] Bai, N.; Xue, K.H.; Huang, J.; Yuan, J.H.; Wang, W.; Mao, G.Q.; Zou, L.; Yang, S.; Lu, H.; Sun, H.; et al. Designing Wake-Up Free Ferroelectric Capacitors Based on the HfO2/ZrO2 Superlattice Structure. Advanced Electronic Materials 2022, 9.

Round 2

Reviewer 1 Report

(1) The paper reports improvement of endurance and reduction of leakage current in ferroelectric Hf-Zr-O (HZO) by inserting a laminated-structure interlayer, which is interesting and timely. In the revised manuscript (v2), the data are compared to those of fully laminated structure, showing clear advantage of the interlayer insertion. The paper reports degradation of Pr for the fully laminated structure and the explanation sounds reasonable. On the other hand, some papers reported “enhancement” of remanent polarization for the fully laminated structure when the sublayer thickness is 1 nm. (see ref.28 and Y. Peng, et al., IEEE Electron Device Lett., 43, 216 (2022)) It would be nice to add comments on the discrepancy between the present results and previous observation of Pr enhancement.

(2) The authors explain the mechanism of Pr reduction in the laminated structures due to the phase fraction decrease of the O-phase compared to the M- and T-phases. Can such phase fraction changes be detected by XRD or TEM analysis? It would be nice if evidence for this explanation could be provided. The reviewer thinks it is good to show XRD patterns for the presented three samples, or please add some comments on the XRD patterns.

Author Response

Manuscript ID: nanomaterials-2369888

Authors’ responses to the reviewer’s comments

Reply to Reviewers’ comments on the manuscript submitted to nanomaterials-2369888, entitled “Improved Endurance of Ferroelectric Hf0.5Zr0.5O2 Using Laminated-Structure Interlayer” by Meiwen Chen et al. (nanomaterials-2369888). We are sincerely grateful to all reviewers, and their constructive comments will guide us to improve the quality of our manuscript. Based on the comments received from reviewers, our group had several in-depth discussions, and have significantly revised the paper according to the constructive comments. The detailed point-by-point responses to the reviewers’ comments are summarized as follows. All the revised text, data, and notes in the manuscript are highlighted in yellow.

===========================

Comments from Reviewer 1#

Comment 1: The paper reports improvement of endurance and reduction of leakage current in ferroelectric Hf-Zr-O (HZO) by inserting a laminated-structure interlayer, which is interesting and timely. In the revised manuscript (v2), the data are compared to those of fully laminated structure, showing clear advantage of the interlayer insertion. The paper reports degradation of Pr for the fully laminated structure and the explanation sounds reasonable. On the other hand, some papers reported “enhancement” of remanent polarization for the fully laminated structure when the sublayer thickness is 1 nm. (see ref.28 and Y. Peng, et al., IEEE Electron Device Lett., 43, 216 (2022)) It would be nice to add comments on the discrepancy between the present results and previous observation of Pr enhancement.

Our response: Thank you for your comments. The enhancement of remanent polarization for fully laminated structure with 1nm-thick sublayer in ref.28. The performance differences may be due to different deposition methods. According to the letter, the starting layer of the device is HfO2. In this article, the starting layers of all three types of capacitors are ZrO2. And we add comments as follow: “It is reported that the HfO2-starting laminated structure exhibited higher remanent polarization and optimal remanent polarization was achieved in thicker sublayer about 1nm[1,2]. The ZrO2-starting laminated structure shows lower remanent polarization and the remanent polarization decreases with deposition cycles increasing[3]. However, it also reported that ZrO2 nucleation layer could stabilize remanent polarization of HZO ferroelectric thin films during field cycling[4].”

Corresponding change in manuscript: Yes

Location of Change:

Section: Introduction

Page 2, line 74-79.

--------------------------

[1] Lehninger, D.; Prabhu, A.; Sünbül, A.; Ali, T.; Schöne, F.; Kämpfe, T.; Biedermann, K.; Roy, L.; Seidel, K.; Lederer, M. Ferroelectric [HfO2/ZrO2] Superlattices with Enhanced Polarization, Tailored Coercive Field, and Improved High Temperature Reliability. Advanced Physics Research, 2023.

[2] Peng, Y.; Xiao, W.; Liu, Y.; Jin, C.; Deng, X.; Zhang, Y.; Liu, F.; Zheng, Y.; Cheng, Y.; Chen, B. HfO2-ZrO2 superlattice ferroelectric capacitor with improved endurance performance and higher fatigue recovery capability. 2021, 43, 216-219.

[3] Park, M.H.; Kim, H.J.; Lee, G.; Park, J.; Lee, Y.H.; Kim, Y.J.; Moon, T.; Kim, K.D.; Hyun, S.D.; Park, H.W. A comprehensive study on the mechanism of ferroelectric phase formation in hafnia-zirconia nanolaminates and superlattices. 2019, 6, 041403.

[4] Jiang, P.; Wei, W.; Yang, Y.; Wang, Y.; Xu, Y.; Tai, L.; Yuan, P.; Chen, Y.; Gao, Z.; Gong, T. Stabilizing Remanent Polarization during Cycling in HZO‐Based Ferroelectric Device by Prolonging Wake‐up Period. 2022, 8, 2100662.

Comment 2: The authors explain the mechanism of Pr reduction in the laminated structures due to the phase fraction decrease of the O-phase compared to the M- and T-phases. Can such phase fraction changes be detected by XRD or TEM analysis? It would be nice if evidence for this explanation could be provided. The reviewer thinks it is good to show XRD patterns for the presented three samples, or please add some comments on the XRD patterns.

Our response: Thank you for your comments. We have done the XRD for HZO1/1 and HZO1/5/1 samples, as shown in Figure R1. However, due to the small difference of the Pr value between the HZO1/1 and HZO1/5/1 capacitors, XRD plots do not particularly clearly show the difference of the O phase and the M phase intensity between the two capacitors. And we have also tried to use the Fourier transform on TEM images to analyze the existence of O/T/M phases, but there are many grain coincidences in the TEM images. In order to obtain more information about the O/T/M phase, we need to prepare more samples and need more detailed STEM experiments. We will conduct this experiment in the next work.

Figure R1. XRD images of HZO1/1 and HZO1/5/1 capacitors.

Corresponding change in manuscript: No

Reviewer 2 Report

In Figure 1(e), it is still unclear whether the lattice arrangement is regular or an incoherent lattice is formed. I recommend at least enlarging the figure image inserted as an inset to a new Figure 1(f) and providing a clear indication of what has become a lattice arrangement as described by the authors. It could be very misleading to suggest that the interlayer effect can be induced by an arrangement of incoherent and inhomogeneous lattices. If the lattice matching effect is simply induced by the interlayer effect, such a remarkable structural effect, the clear and obvious evidence in the view of nanomaterials science for this is essential. The revised text also incorrectly describes the description of Figure 1(d) and the inset figure. 

I would also like to see additional analytical results and XRD data not included by the authors as supporting figures. 

Minor typos, fonts, and style issues should be corrected.

Author Response

Manuscript ID: nanomaterials-2369888

Authors’ responses to the reviewer’s comments

Reply to Reviewers’ comments on the manuscript submitted to nanomaterials-2369888, entitled “Improved Endurance of Ferroelectric Hf0.5Zr0.5O2 Using Laminated-Structure Interlayer” by Meiwen Chen et al. (nanomaterials-2369888). We are sincerely grateful to all reviewers, and their constructive comments will guide us to improve the quality of our manuscript. Based on the comments received from reviewers, our group had several in-depth discussions, and have significantly revised the paper according to the constructive comments. The detailed point-by-point responses to the reviewers’ comments are summarized as follows. All the revised text, data, and notes in the manuscript are highlighted in yellow.

===========================

Comments from Reviewer 2#

Comment 1: In Figure 1(e), it is still unclear whether the lattice arrangement is regular or an incoherent lattice is formed. I recommend at least enlarging the figure image inserted as an inset to a new Figure 1(f) and providing a clear indication of what has become a lattice arrangement as described by the authors. It could be very misleading to suggest that the interlayer effect can be induced by an arrangement of incoherent and inhomogeneous lattices. If the lattice matching effect is simply induced by the interlayer effect, such a remarkable structural effect, the clear and obvious evidence in the view of nanomaterials science for this is essential. The revised text also incorrectly describes the description of Figure 1(d) and the inset figure.

I would also like to see additional analytical results and XRD data not included by the authors as supporting figures.

Our response: Thank you for your comment. We are sorry that our conjecture on lattice arrangement based on stratification phenomena and electrical test results may not be rigorous enough, and we do not have sufficient evidence to support this conjecture. So, we removed the description from the article. And we have enlarged the inserted image as Figure 1(f) and modified the description in the article, as shown in Figure R1. And it is difficult to show the difference of the O phase and the M phase intensity between the two capacitors through XRD characterization for the small difference of the Pr value between the HZO1/1 and HZO1/5/1 capacitors. We hope to adjust the thickness of the laminated structure and complete the characterization in the next work.

Figure R1. Cross-sectional TEM images of (d) the capacitor HZO1/1 and (e) the capacitor HZO1/5/1. (f) Magnified image extracted in (e).

Corresponding change in manuscript: Yes

Location of Change:

Section: Introduction

Page 3, line 128-129.